# Seepage Comprehensive Evaluation of Concrete Dam Based on Grey Cluster Analysis

**Junjie Li [1] , Xudong Chen [1,\*], Chongshi Gu [2,3] and Zhongyan Huo [4]**

[1] College of Water Conservancy & Environmental Engineering, Zhengzhou University, Zhengzhou 450001, China
[2] State Key Laboratory of Hydro-Water Resources and Hydraulic Engineering, Hohai University, Nanjing 210098, China
[3] College of Water Conservancy and Hydropower Engineering, Hohai University, Nanjing 210098, China
[4] School of Port and Transportation Engineering, Zhejiang Ocean University, Zhoushan 316000, China
\* Correspondence: chenxudong@zzu.edu.cn; Tel.: +86-183-3630-2520

**Abstract:** Most concrete dams have seepage problems to some degree, so it is a common strategy to maintain ongoing monitoring and take timely repair measures. In order to grasp the real operation state of dam seepage, it is vital to analyze the measured data of each monitoring indicator and establish an appropriate prediction equation. However, dam seepage states under the load and environmental influences are very complicated, involving various monitoring indicators and multiple monitoring points of each indicator. For the purpose of maintaining the temporal continuity and spatial correlation of monitoring objects, this paper used a multi-indicator grey clustering analysis model to explore the grey correlation among various indicators, and realized a comprehensive evaluation of a dam seepage state by computation of the clustering coefficient. The case study shows that the proposed method can be successfully applied to the health monitoring of concrete dam seepage.

**Keywords:** dam seepage; comprehensive indicator system; seepage monitoring model; grey clustering analysis

---

## 1. Introduction

After a dam is completed and impounded, once the seepage factors (seepage head, seepage gradient, etc.) exceed the allowable value, the dam will suffer from seepage damage. The related statistic shows that dam failures caused by seepage problems account for 30% to 40% of the total number of failures, second only to flood overtopping [1]. Therefore, it is extremely important to carry out comprehensive monitoring of dam seepage, which is the most effective way to monitor operating conditions, find unsafe factors, and prevent problems before accidents occur [2]. However, dam seepage states are affected by many external factors, such as water level, rainfall, temperature of dam concrete, and time effects [3], which makes comprehensive seepage evaluation very complicated. It is essential to find the variation rules of load set with dam seepage, as well as exploring the relationship between them. The reasonable analysis method is to select the factors and expressions of the seepage monitoring variables by deterministic function [4] and statistical correlation [5] according to the basic principle of seepage flow, and then calculate the coefficients of the model with mathematical statistics [6] based on monitoring data. By using mathematical expressions between independent variables and dependent variables, the monitoring data can be fitted and the variation law of seepage in the near future can be predicted.

To date, studies [7–9] on the evaluation of dam seepage state have been mainly aimed at a single evaluation indicator, which can only reflect partial information of the dam's seepage state and will therefore lead to inaccuracy and incompleteness of the evaluation results. In practical engineering, the seepage state of concrete dams should be reflected by several indicators (dam seepage pressure, dam leakage, and seepage around dam) [10], and multiple monitoring points of each indicator distributed at different elevations in different dam sections. It is difficult to evaluate dam seepage states directly according to the information of the monitoring points, so it is impossible to know whether there is an abnormal phenomenon of seepage during the operation of the dam. In order to realize the comprehensive evaluation of dam seepage safety, it is necessary to fuse the information of multiple indicators with multiple monitoring points and explore the correlations among various indicators, which is the main purpose and novelty of this article. There are many uncertain problems [11] in the process of multi-information fusion, including incomplete information, incomplete data, incommensurability of multiple indicators, etc. The commonly used ladder fusion methods mainly include the following: set pair analysis theory [12], neural network [13], risk reliability combined theory [14], matter element extension model [9], cluster fusion diagnosis model [15], cloud model [16], defuzzification method [17], etc. However, most concrete dams built many years ago have limitations of monitoring conditions and the randomness of manual monitoring, resulting in the problem of short monitoring data sequences and inaccurate data. As a method of dealing with uncertain systems, small samples, and poor information, grey clustering analysis theory can extract valuable information by generating and mining [18], which is relatively suitable for the comprehensive evaluation of dam seepage safety. Combining this theory with a seepage monitoring model, a multi-indicator grey clustering comprehensive analysis model of dam seepage has been constructed, which integrates the monitoring data of each indicator according to defined categories, and generates a whitenization weight function value [19] and comprehensive clustering coefficient. In this process, the model outputs the grey correlation degree [20] between different indicators and analyzes the change tendency of the seepage state, and then divides the seepage state into different grey classes according to the fluctuations of the monitoring data. This can give the staff a degree of judgment on the seepage state and provide beneficial reference values for dam maintenance, so that we can visually find the parts with abnormal seepage flow to prevent the occurrence of dangerous accidents.

This article is organized as follows. Section 2 introduces the seepage monitoring model of concrete dams, which includes the establishment of a comprehensive indicator system of dam seepage state and a statistical model, as well as the standard of grading. The grey clustering analysis theory is described in Section 3.1, and a multi-indicator grey clustering comprehensive analysis model is established in Section 3.2. The application of engineering cases is introduced in Section 4. Concluding remarks complete the paper in Section 5.

## 2. Seepage Monitoring Model of Concrete Dams

### 2.1. The Comprehensive Indicator System of Dam Seepage State

The evaluation indicators of dam seepage state constitute a multi-level and multi-objective comprehensive indicator system, which is mainly drawn from the following three aspects: (1) Dam seepage pressure is composed of uplift pressure of the dam foundation and seepage pressure of the dam body. Both are caused by seepage water and have certain influences on dam stability, deformation, and stress; (2) Dam leakage causes the water to take the fine particles out of the dam body and form a seepage passage, which endangers the stability of the dam; (3) Upstream impoundment can not only seep through the dam body and foundation, but also seep downward around the bank slopes at both ends of the dam. Considering the temporal continuity and spatial correlation [21] of the monitoring points, each indicator is jointly monitored by several monitoring points for a long time series. With reference to the monitoring data and the dam seepage monitoring research results [10,22,23], the comprehensive indicator system of concrete dam seepage state is shown in Figure 1.

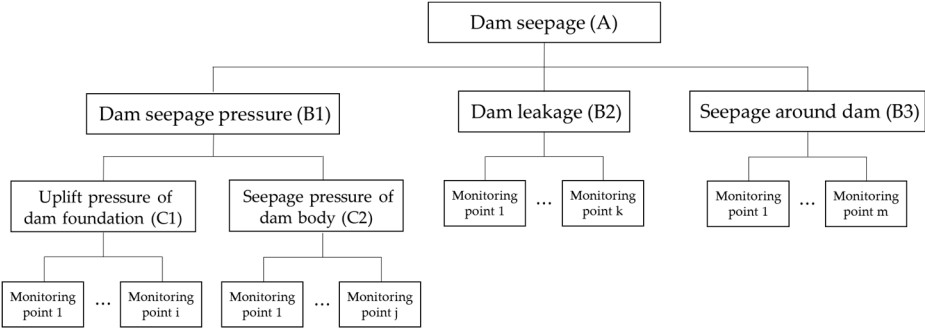

**Figure 1.** Comprehensive evaluation indicator system of concrete dam seepage.

*2.2. Seepage Monitoring Model*

According to dam safety monitoring theory [24,25], concrete dam seepage is composed of four components: water level component $y_H$, rainfall component $y_P$, temperature component $y_T$, and time effect component $y_\theta$. It can be expressed as follows:

$$y_t = y_H + y_P + y_T + y_\theta = f(H, P, T, \theta) \tag{1}$$

The four components are described below.

2.2.1. Water Level Component $y_H$

Through analysis of test data and seepage theory [4] to deduce the expression form of the water pressure effect, there is a certain hysteresis and linear dependence between the change of upstream water level with uplift pressure of dam foundation and seepage around dam, so the water level component is usually expressed as follows:

$$y_H = \sum_{i=1}^{5} a_i(h_i - h_{0i}) \tag{2}$$

where $a_i$ ($i$ = 1–5) is the regression coefficient of the water level component; $h_i$ ($i$ = 1–5) is the upstream water level of the monitoring day, 1 day before the monitoring day, average upstream water level of 2 to 4 days before the monitoring day, 5 to 15 days before the monitoring day, and 16 to 30 days before the monitoring day. $h_{0i}$ ($i$ = 1–5) is the average upstream water level corresponding to each of the above periods on the initial monitoring day.

The permeability coefficient of concrete is small, the aggregate gradation is different from the particle material, and the permeability equation is complex. Therefore, besides the first power of upstream water depth, the water level component may also be related to the higher power of upstream water depth, so it is taken to the fourth power to simplify simulation of the influence of upstream water level on the seepage pressure of the dam body, with reference to the literature [26]. The water level component of the seepage pressure of the dam body is often expressed as follows:

$$y_H = \sum_{i=1}^{4} a_i\left(h_{u1}{}^i - h_{u0}{}^i\right) \tag{3}$$

where $a_i$ ($i$ = 1–4) is the regression coefficient of the water level component; $h_{u1}$ is the upstream water depth of the monitoring day; and $h_{u0}$ is the upstream water depth of the initial monitoring day.

Based on the derivation of seepage theory [26], the leakage of the bank slope and river bed dam section is obtained. The dam leakage is related to the first and second power of upstream water depth.

At the same time, the hysteretic effect of reservoir water level on the dam leakage is considered. The water level component of dam leakage is often expressed as follows:

$$y_H = \sum_{i=1}^{2} a_i \left( h_{u1}{}^i - h_{u0}{}^i \right) + \sum_{i=3}^{7} a_i \left( h_{u(i-2)} - h_{u0} \right) \tag{4}$$

where $a_i$ ($i$ = 1–7) are the regression coefficients of the water level component; $h_{u(i-2)}$ are the upstream water level of the monitoring day, 1 day before the monitoring day, average upstream water level of 2 to 4 days before the monitoring day, 5 to 15 days before the monitoring day, and 16 to 30 days before the monitoring day.

### 2.2.2. Rainfall Component $y_P$

The seepage pressure of the dam foundations and both sides of dam is affected by groundwater, which is mainly caused by rainfall in addition to the reservoir water level. Rainfall can seep into the dam body, foundation rock, and bank slope, which will influence the dam seepage. The relationship between rainfall and groundwater level is complex, and is related to rainfall and rainfall pattern, infiltration conditions, topography, and geological conditions, and there is a certain hysteresis. The average rainfall was used as a factor to simplify the simulated rainfall component, as shown below:

$$y_P = \sum_{i=1}^{4} b_i (P_i - P_{0i}) \tag{5}$$

where $P_i$ ($i$ = 1–4) is the rainfall of the monitoring day, 1 day before the monitoring day, the average rainfall of 2 to 4 days before the monitoring day, and 5 to 8 days before the monitoring day; $P_{0i}$ ($i$ = 1–4) is the average rainfall corresponding to each of the above periods on the initial monitoring day; and $b_i$ ($i$ = 1–4) is the regression coefficient of the rainfall component.

### 2.2.3. Temperature Component $y_T$

The temperature component is the variation of seepage caused by the temperature change of the dam concrete and foundation rock. Thermal expansion will decrease the crack gaps, enhance the impermeability, and then relieve seepage, while cooling shrinkage will increase the crack gaps, undermine the impermeability, and then intensify seepage. The temperature of the dam body and foundation rock varies periodically with atmospheric temperature, which can be expressed with a periodic function. Considering the linear relationship between seepage and temperature, the multi-period harmonic was chosen as the factor to represent the temperature component:

$$y_T = \sum_{i=1}^{2} \left[ c_{1i} \left( sin\frac{2\pi it}{365} - sin\frac{2\pi it_0}{365} \right) + c_{2i} \left( cos\frac{2\pi it}{365} - cos\frac{2\pi it_0}{365} \right) \right] \tag{6}$$

where $c_{1i}$ and $c_{2i}$ ($i$ = 1–2) are the regression coefficients, respectively; $t$ is the cumulative number of days from the monitoring day to the initial monitoring day; and $t_0$ is the cumulative number of days from the first monitoring day of the data sequence taken by the modeling to the initial monitoring day.

### 2.2.4. Time Effect Component $y_\theta$

The time effect component is the variation of seepage caused by the change of time. The main factors affecting the time effect component are the viscous flow of the dam concrete, and creep of the rock foundation, which are often slow and undetectable, but the accumulation of such slow deformation causes the stress of the foundation to adjust with time and has a great impact on the distribution of dam foundation cracks. The influence of these factors on seepage flow is a slow time effect process, and it is difficult to obtain accurate expressions by theoretical analysis at present. Typically, variation

of the time effect component usually changes sharply at the beginning, and gradually turns stable during the later period. A formula with a combination of two empirical formulas (linear function and logarithmic function) represents the mathematical model of the general variation law of the time effect component, which can be expressed as follows:

$$y_\theta = d_1(\theta - \theta_0) + d_2(ln\theta - ln\theta_0) \tag{7}$$

where $d_1$ and $d_2$ are regression coefficients; $\theta$ is the number of days from the initial monitoring day divided by 100; and $\theta_0$ is the number of days from the first monitoring day of the data sequence divided by 100, in other words, $\theta = 0.01t$, $\theta_0 = 0.01t_0$.

In summary, the seepage monitoring model can be expressed as follows:

$$
\begin{aligned}
\hat{y}_t = a_0 \;\; & + \sum_{i=1}^{5} a_i(h_i - h_{0i}) or \sum_{i=1}^{4} a_i\left(h_{u1}{}^i - h_{u0}{}^i\right) or \left[\sum_{i=1}^{2} a_i\left(h_{u1}{}^i - h_{u0}{}^i\right) + \sum_{i=3}^{7} a_i\left(h_{u(i-2)} - h_{u0}\right)\right] \\
& + \sum_{i=1}^{4} b_i(P_i - P_{0i}) \\
& + \sum_{i=1}^{2} \left[c_{1i}(sin\tfrac{2\pi it}{365} - sin\tfrac{2\pi it_0}{365}) + c_{2i}(cos\tfrac{2\pi it}{365} - cos\tfrac{2\pi it_0}{365})\right] \\
& + d_1(\theta - \theta_0) + d_2(ln\theta - ln\theta_0)
\end{aligned}
\tag{8}
$$

where $\hat{y}_t$ is the fitting value of the seepage monitoring data; and $a_0$ is a constant.

Seepage monitoring values $y_t$ are variable, and the fitting values $\hat{y}_t$ obtained by regression analysis will deviate from the monitoring value to a certain extent. After establishment of the seepage monitoring model, the accuracy of the regression fitting can be measured by residual standard deviation ($S$) and the complex correlation coefficient ($R$).

$$S = \sqrt{\frac{\sum_{t=1}^{n}(y_t - \hat{y}_t)^2}{n - k - 1}} \tag{9}$$

$$R = \sqrt{\frac{\sum_{t=1}^{n}(\hat{y}_t - \overline{y_t})^2}{\sum_{t=1}^{n}(y_t - \overline{y_t})^2}} \tag{10}$$

where $n$ is the number of monitoring data, $k$ is the number of independent variables, and $\overline{y_t}$ is the average value of monitoring data.

### 2.3. Safety Class Division Standard

The working state of seepage can be graded with the Pauta criterion [27]. It is considered that the fitting curve calculated by the model is a reasonable value, and the confidence interval class is determined according to a certain probability. The data in this interval are considered to be a reasonable fluctuation of seepage. Any error exceeding the range is not a random error but a gross error, and the data containing this error are determined to be in an abnormal state. The interval distributions of three safety classes of the dependent variables are shown in Figure 2.

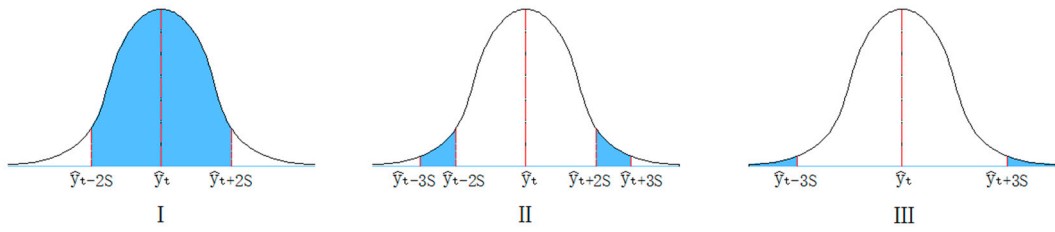

**Figure 2.** Interval distribution diagram of three safety classes of dependent variables.

When $\left|y_t - \hat{y}_t\right| \leq 2S$, it belongs to class I, indicating that the working state is normal;

When $2S < \left|y_t - \hat{y}_t\right| \leq 3S$, it belongs to class II, indicating that the working state is basically normal. If there is an obvious tendency variation, it indicates that the working state is abnormal;

When $\left|y_t - \hat{y}_t\right| > 3S$, it belongs to class III, indicating that the working state is dangerous.

The tendency variation is mainly reflected in the change of the time effect component. Positive and negative values of $dy_\theta / dt$ indicate the increase and decrease of the time effect component, and the size of $d^2 y_\theta / dt^2$ indicates the rate of increase and decrease of the time effect component; $dy_\theta / dt$ and $d^2 y_\theta / dt^2$ can be used to achieve judgment of the variation tendency [28].

## 3. Multi-Indicator Grey Clustering Comprehensive Analysis

To deal with uncertainty problems during comprehensive evaluation with multi-indicators, grey system theory is often adopted, which is used to solve the problem of uncertain systems, small samples, and poor information [29]. The grey system studies the structure and function of a "black-grey-white" box [30] through the organic relations and change rules among the object, element, and environment, as shown in Figure 3.

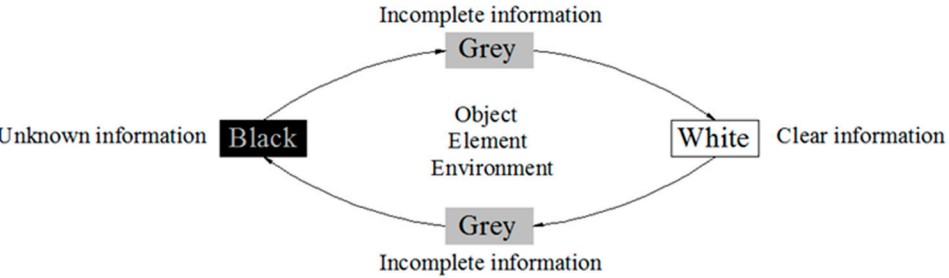

**Figure 3.** Concept map of the grey system.

### 3.1. Grey Clustering Analysis Theory

Grey clustering is a method of dividing monitoring indicators or monitoring objects into several definable categories according to the grey correlation matrix or the grey whitenization weight function. A cluster can be seen as a collection of monitoring objects belonging to the same class. This theory extracts valuable information by generating and mining some known information to calculate the grey correlation between different indicators and simplify complex systems [31].

#### 3.1.1. Basic Model

Suppose that $x_{ij}$ $(i = 1, 2, \cdots, n; j = 1, 2, \cdots, m)$ is the monitoring value of object $i$ about indicator $j$, and $f_j^k(\cdot)$ $(j = 1, 2, \cdots, m; k = 1, 2, \cdots, s)$ is the definite weighted function of the class $k$ about indicator $j$. The grey clustering coefficient of the object $i$ belonging to the grey class $k$ is:

$$\sigma_i^k = \sum_{j=1}^{m} f_j^k(x_{ij}) \cdot \eta_j \tag{11}$$

where $\eta_j$ is the weight of grey class $k$ about indicator $j$.

#### 3.1.2. Modeling Steps

(1) According to the evaluation requirements, the indicator $j$ is divided into $s$ grey classes, and the range of values of each indicator is also divided into $s$ grey classes. For example, the value range $[a_1, a_{s+1}]$ of the indicator $j$ is divided into $s$ small intervals:

$$[a_1, a_2], [a_2, a_3], \cdots, [a_{s-1}, a_s], [a_s, a_{s+1}]$$

(2) Calculation of the geometric midpoint between each small interval, $\lambda_k = (a_k + a_{k+1})/2$, $k = 1, 2, \cdots, s$;

(3) Assume that the whitenization weight function value of $\lambda_k$ belonging to the grey class $k$ is 1. Connect the point $(\lambda_k, 1)$ and the geometric midpoint $(\lambda_{k-1}, 0)$ of the grey class $k - 1$ to obtain the triangular definite weighted functions $f_j^k(\cdot)$, $j = 1, 2, \cdots, m; k = 1, 2, \cdots, s$. For $f_j^1(\cdot)$ and $f_j^s(\cdot)$, the indicator $j$ can be extended to the left and right to $a_0$, $a_{s+2}$ respectively. For a monitoring value $x$ of the indicator $j$, it can be obtained as follows:

$$f_j^k(x) = \begin{cases} 0, x \notin [a_{k-1}, a_{k+2}] \\ \frac{x - a_{k-1}}{\lambda_k - a_{k-1}}, x \in [a_{k-1}, \lambda_k] \\ \frac{a_{k+2} - x}{a_{k+2} - \lambda_k}, x \in [\lambda_k, a_{k+2}] \end{cases} \tag{12}$$

The membership degree $f_j^k(x)$ of the grey class $k$ ($k = 1, 2, \cdots, s$) is calculated, as shown in Figure 4 (The letters in the figure are consistent with the previous variables).

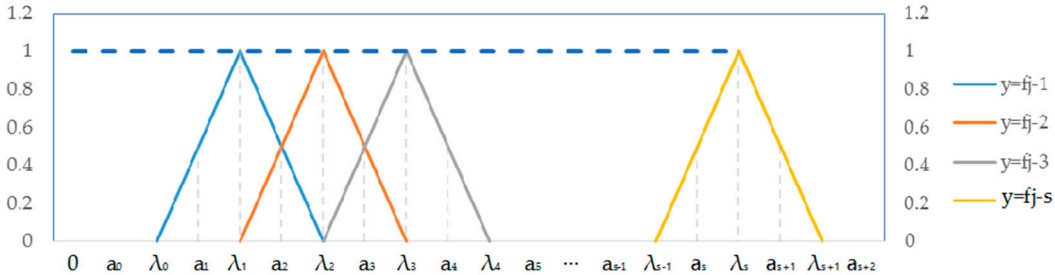

**Figure 4.** Schematic diagram of the endpoint triangle whitenization weight function.

(4) In the process of multi-indicator grey clustering comprehensive analysis, machine learning is used without human intervention. Therefore, the objective weighting method is used to determine the weights of indicators. Entropy weight method is a method of calculating indicator weight based on comprehensive consideration of the information provided by various factors [32]. It calculates the information entropy of each indicator to determine its difference, and then determines the importance of the indicator in the comprehensive evaluation.

If there are $n$ objects and $m$ indicators, the prototype monitoring indicator values are normalized to obtain the matrix $S_{ij} = [s_{ij}]$, and $s_{ij}$ is the normalized value of the indicator $j$ of the object $i$. According to the information entropy definition, the information entropy of the indicator $j$ is:

$$P_j = -q \sum_{i=1}^{n} s_{ij} ln s_{ij} \quad (q = \frac{1}{ln n} \; j = 1, \cdots, m) \tag{13}$$

The entropy weight of indicator $j$ is:

$$e_j = \frac{1 - P_i}{m - \sum_{j=1}^{m} P_i} \quad (0 \leq e_j \leq 1, \sum_{j=1}^{m} e_j = 1) \tag{14}$$

An indicator weight matrix $A = (e_1, e_2, \cdots, e_j)$ of the dam safety comprehensive early warning system based on the entropy weight method is constructed.

(5) The comprehensive clustering coefficients of the object $i$ ($i = 1, 2, \cdots, n$) about the grey class $k$ ($k = 1, 2, \cdots, s$) are calculated, as shown in Equation (11).

(6) From $\max\limits_{1 \leq k \leq s} \{\sigma_i^k\} = \sigma_i^{k*}$, judge that object $i$ belongs to grey class $k^*$; when there are multiple objects belonging to the grey class $k^*$, the superiority or inferiority of the objects belonging to the grey class $k^*$ can be further determined according to the size of the comprehensive clustering coefficient.

### 3.2. Multi-Indicator Grey Clustering Comprehensive Analysis of Concrete Dam Seepage

The multi-indicator grey clustering comprehensive analysis model is here combined with the seepage monitoring model and grey cluster analysis theory, which can realize a comprehensive evaluation of seepage state. The first step of the model is to check the data of the monitoring indicators and environmental variables, which are collected by the dam seepage monitoring system. The seepage monitoring model is used to analyze and fit the monitoring data of each indicator, and the trend of each indicator is then predicted through mathematical expression of the model. Finally, grey clustering analysis theory is used to centralize multiple indicators and realize the comprehensive evaluation of seepage state. The overall framework of steps is shown in Figure 5.

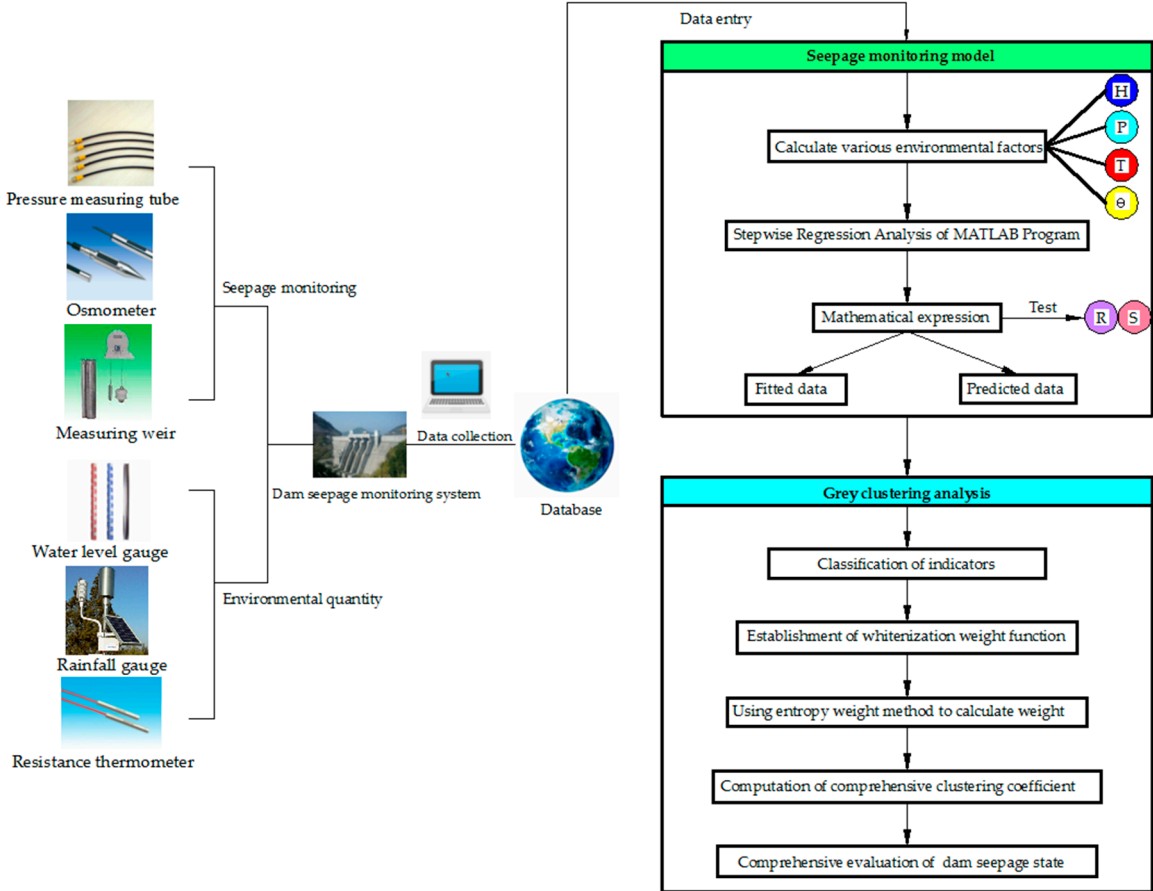

**Figure 5.** Overall framework of the multi-indicator grey clustering comprehensive analysis of dam seepage.

## 4. Case Study

The Cotton Beach Waterpower Project is located in Fujian Province, China. The project belongs to class I. It is a large, multi-purpose project with the main purpose of power generation, and other functions including flood control, navigation, and aquaculture. The maximum dam height is 111.0 m, the dam crest elevation is 179.0 m, and the total length of the dam crest is 308.5 m. The normal water level of the reservoir is 173 m, and the storage capacity is 1.122 billion m$^3$. The check flood level is 177.80 m, and the total storage capacity is 2.035 billion m$^3$. The dead water level is 146 m. The dam has seven dam sections: the first and second dam sections are left bank retaining dam sections, the third and fourth dam sections are overflow dam sections, and the fifth, sixth, and seventh dam sections are right bank retaining dam sections. The installed capacity is 600,000 kW. In order to ensure the seepage state of the dam, a comprehensive monitoring project is arranged inside of the main buildings. The information of monitoring points is shown in Table 1, and the location distribution of the monitoring



points is shown in Figure 6. The monitoring points of seepage around the dam are arranged in the seepage observation holes on both sides of the rock mass, and there are four monitoring points on both the left and right sides of the dam (L1, R1 are in front of the dam curtain and the others are behind the dam curtain).

**Table 1.** The information of monitoring points.

| Monitoring Object | Number of Monitoring Points | Monitoring Instrument |
| --- | --- | --- |
| Dam seepage pressure | UP1, UP3, UP5, UP7, UP9, UP11, UP13, UP15, UP16 | Pressure measuring tube/osmometer |
| Dam leakage | WE1, WE2, WE3, WE4, WE7, WE10 | Measuring weir/osmometer |
| Seepage around dam | L1, L3, L5, L7, R1, R3, R5, R6 | Measuring weir/osmometer |

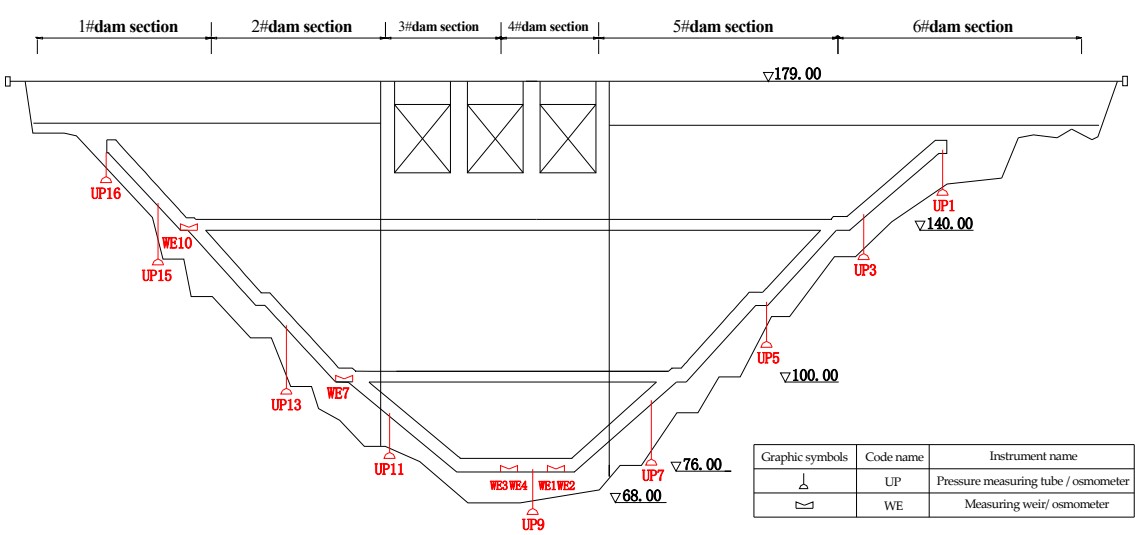

**Figure 6.** Location distribution map of the dam seepage monitoring points.

*4.1. Calculation of Seepage Monitoring Model*

Step 1: Collection of data from monitoring indicators and environmental factors from January 1, 2004 to October 31, 2008 through the dam seepage monitoring system.

Step 2: Substitution of these data into Equation (8) of the seepage monitoring model. By stepwise regression analysis, the regression coefficients were solved, and the expression of the seepage monitoring model was obtained.

Step 3: The data of environmental factors from November 1, 2008 to December 31, 2008 were substituted into the calculated seepage monitoring model, and the model output value of the monitoring points was predicted. Figure 7 shows the variation curves between the monitoring values and the model output values of the monitoring points.

From the variation curves and the multiple correlation coefficients of the monitoring values and the model output values, it can be seen that the seepage monitoring model fit the monitoring sequence well, the trends were consistent, and the values were similar. The model values can be used as reasonable values of the monitoring sequence to discriminate the existence of the abnormal values of the large offset data in the monitoring sequence, so as to realize class division of the working states of the seepage indicators, according to the differences of the confidence probability. And the weight of each monitoring point is calculated with the entropy weight method. The classification and weight of each indicator are shown in Tables 2–4.

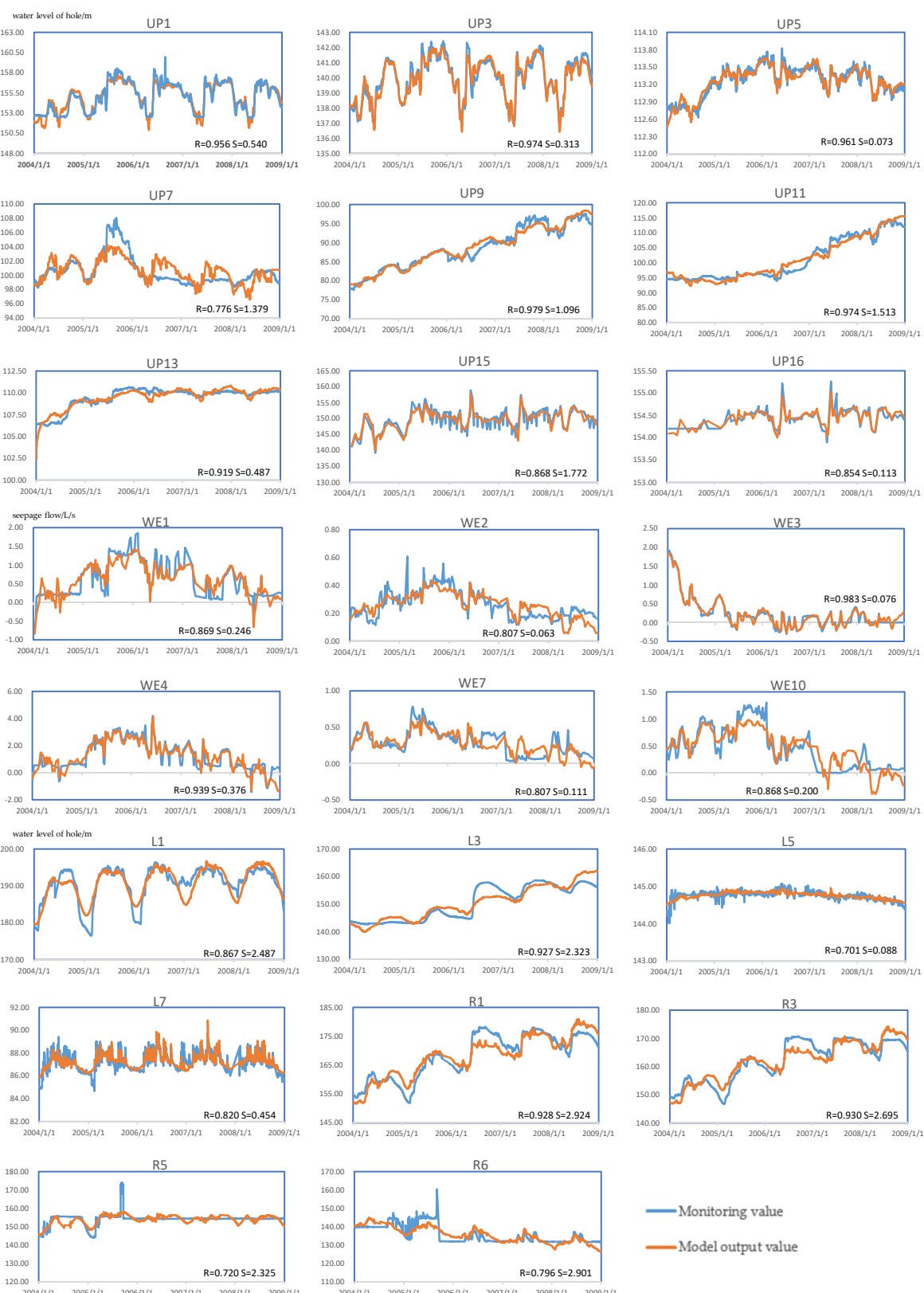

**Figure 7.** The variation curves between the monitoring values and the mode output values.

**Table 2.** The classification results and weights of dam seepage pressure.

| Class | UP1 | UP3 | UP5 | UP7 | UP9 | UP11 | UP13 | UP15 | UP16 |
|---|---|---|---|---|---|---|---|---|---|
| Class I | [0, 1.08] | [0, 0.63] | [0, 0.15] | [0, 2.76] | [0, 2.19] | [0, 3.03] | [0, 0.97] | [0, 3.54] | [0, 0.23] |
| Class II | (1.08, 1.62] | (0.63, 0.94] | (0.15, 0.22] | (2.76, 4.14] | (2.19, 3.29] | (3.03, 4.54] | (0.97, 1.46] | (3.54, 5.32] | (0.23, 0.34] |
| Class III | (1.62, +∞) | (0.94, +∞) | (0.22, +∞) | (4.14, +∞) | (3.29, +∞) | (4.54, +∞) | (1.46, +∞) | (5.32, +∞) | (0.34, +∞) |
| Information entropy | 61.902 | 60.892 | 66.481 | 60.338 | 57.992 | 54.404 | 36.561 | 9.097 | 9.094 |
| Weight | 0.071 | 0.069 | 0.076 | 0.069 | 0.066 | 0.062 | 0.041 | 0.009 | 0.009 |

**Table 3.** The classification results and weights of dam leakage.

| Class | WE1 | WE2 | WE3 | WE4 | WE7 | WE10 |
|---|---|---|---|---|---|---|
| Class I | [0, 0.49] | [0, 0.13] | [0, 0.15] | [0, 0.75] | [0, 0.22] | [0, 0.40] |
| Class II | (0.49, 0.74] | (0.13, 0.19] | (0.15, 0.23] | (0.75, 1.13] | (0.22, 0.33] | (0.40, 0.60] |
| Class III | (0.74, +∞) | (0.19, +∞) | (0.23, +∞) | (1.13, +∞) | (0.33, +∞) | (0.60, +∞) |
| Information entropy | 8.464 | 12.763 | 4.495 | 9.217 | 12.511 | 10.045 |
| Weight | 0.009 | 0.014 | 0.005 | 0.010 | 0.013 | 0.010 |

**Table 4.** The classification results and weights of seepage around dam.

| Class | L1 | L3 | L5 | L7 | R1 | R3 | R5 | R6 |
|---|---|---|---|---|---|---|---|---|
| Class I | [0, 4.97] | [0, 4.65] | [0, 0.18] | [0, 0.91] | [0, 5.85] | [0, 5.39] | [0, 4.65] | [0, 5.80] |
| Class II | (4.97, 7.46] | (4.65, 6.97] | (0.18, 0.26] | (0.91, 1.36] | (5.85, 8.77] | (5.39, 8.09] | (4.65, 6.98] | (5.80, 8.70] |
| Class III | (7.46, +∞) | (6.97, +∞) | (0.26, +∞) | (1.36, +∞) | (8.77, +∞) | (8.09, +∞) | (6.98, +∞) | (8.70, +∞) |
| Information entropy | 42.261 | 36.613 | 59.311 | 66.776 | 46.283 | 45.546 | 74.592 | 39.119 |
| Weight | 0.048 | 0.041 | 0.068 | 0.076 | 0.053 | 0.052 | 0.085 | 0.044 |

### 4.2. Comprehensive Clustering Calculation of Multiple Dependent Variables

The whitenization weight function is the basis of grey clustering. The membership function of fuzzy evaluation can reflect the relationships between clustering indicators and grey classes. Since the evaluation factors of monitoring sequence indicators have a lower limit and no upper limit, the upper and lower limit whitenization weight function was used, as shown in Figure 8.

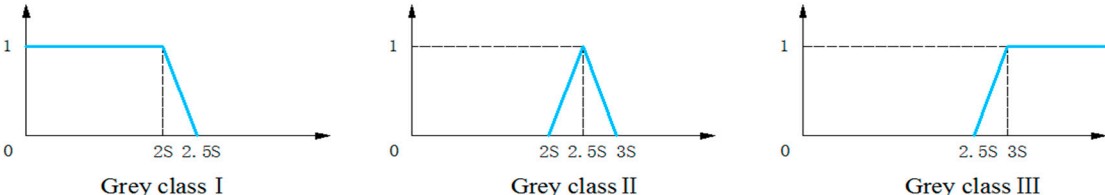

**Figure 8.** Schematic diagram of the definite weighted function.

The multi-indicator grey clustering comprehensive analysis model was used on the period from November 1, 2008 to December 31, 2008 to forecast the change trend of seepage, combining the monitoring data to calculate the $\left|y_t - \hat{y}_t\right|/S$ of each monitoring points and the clustering coefficients of each seepage indicator, as shown in Figure 9.

### 4.3. Results and Discussion

Each indicator has three grey class clustering coefficients; the bigger the clustering coefficient of the grey class, the more likely the working state of the indicator will be in that grey class. The larger the clustering coefficient of grey class I is, the better the seepage state will be. On the contrary, the larger the clustering coefficient of grey class III is, the worse the seepage state will be. The sum of the three grey classes at any time is 1.

From Figure 9, it can be seen that the state of dam seepage pressure was basically in the stage of normal operation of grey class I across these two months, which were both greater than 0.75. After December, the proportion of grey class II increased, reaching a maximum of 0.25, and dangerous conditions of grey class III began to emerge, indicating that the state of dam seepage pressure tends to develop towards the abnormal, which was mainly attributed to the large offset anomaly of monitoring points UP9 and UP11.

The dam leakage is monitored manually, and only once a week, so the monitoring sequence is less than automatic monitoring. It can be seen that the grey class I of dam leakage was basically greater than 0.75, which would indicate normal operation. However, grey class II had an upward trend, and gradually changed to grey class III, with a maximum of 0.23, indicating that the state of dam leakage had a trend of dangerous development, which was mainly attributed to the large offset anomaly of monitoring points WE4 and WE3.

The grey class I of seepage around the dam was generally higher than 0.9, the maximum grey class II reached 0.1, and only monitoring point L3 had a small offset anomaly, indicating that seepage around dam was in normal condition during these two months.

Figure 10 is a comprehensive clustering coefficient histogram of seepage, which takes into account the above three indicators. It shows that the dam seepage was generally in the stage of grey class I during this period, and there was an upward trend of grey class II after December, reaching a maximum of 0.13, indicating that the seepage was in a good state of operation, but there was a small offset anomaly.

**Figure 9.** $\left| y_t - \hat{y}_t \right| / S$ and the clustering coefficient of each seepage indicator.

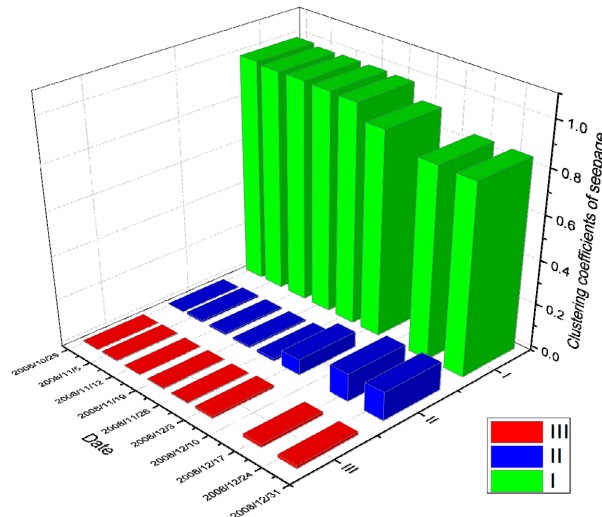

**Figure 10.** Comprehensive clustering coefficient histogram of seepage.

In summary, although dam seepage was overall in normal operation, some abnormal conditions were discovered through the multi-indicator grey clustering comprehensive analysis model. The data of some monitoring points had obvious deviations from the model output values, such as UP9, UP11, WE3, and WE4, which were distributed in sections 3# and 4# of the overflow dam section, indicating that the overflow dam section is the main section where seepage anomaly occurs. Because the overflow dam section sometimes needs continuous drainage, it is easy for water flow to scour the dam and negative pressure to appear on overflow surface, resulting in leakage and intestinal retention of fluid. In order to ensure the normal operation of dam seepage, it is necessary to monitor abnormal parts moving forward to prevent the occurrence of dangerous accidents.

## 5. Conclusions

The purpose of this paper was to construct a multi-indicator grey clustering comprehensive analysis model for seepage safety monitoring of concrete dams, using the seepage monitoring model to predict the change tendency of monitoring points, combined with grey cluster analysis theory to fuse the multiple indicators, so as to realize a comprehensive evaluation of the dam seepage state. The following conclusions can be obtained:

The seepage indicator system is complete and representative, and takes into account the main factors affecting the seepage state of the dam, and each factor includes multiple monitoring points of the whole dam section. The seepage monitoring model adopts different mathematical expressions for different indicators according to the correlation between water level component and indicators. Stepwise regression method is used to retain the environmental factors with high correlation and eliminate the environmental factors with poor correlation, which makes the model fit the monitoring data better. By using the Pauta criterion to classify the indicators according to a certain probability, the problem of random errors in actual monitoring and statistics was eliminated.

Grey clustering analysis theory combined with the seepage monitoring model was applied to the integration of multi-source information monitoring data of dam seepage, which is an interdisciplinary practice. The evaluation results reflect the trend of dam seepage through the continuous change of grey class with time series, which is a major improvement from static state to dynamic state evaluation. There are many uncertainties in the fusion of multi-index and multi-information; grey clustering analysis theory can find the grey correlations among the indicators, mine useful information from limited data, and divide observation objects into several definable categories with a whitenization weight function. This combination of different factors can simplify the complex system and is very suitable for comprehensive evaluation of dam seepage. Through the calculation of an engineering case, the validity and availability of the method was verified. In practical engineering applications, when the calculation results are generally shown as grey class I, it indicated that the dam seepage was in the normal operation stage. However, attention should be paid to the changing trends of the other grey classes. If there is a tendency variation of offset anomaly, it indicates that the dam seepage state is deteriorating gradually. When the calculation results reach grey class II, preventive measures must be taken for abnormal parts, such as drainage decompression, curtain grouting, concrete filling, etc. The seepage state of the dam is expressed quantitatively, which is convenient for the daily maintenance of the dam and the early warning of the dangerous situation. This method can also be applied to other engineering fields. As long as data with multiple evaluation indicators are available, valuable information can be extracted through the multi-indicator grey clustering comprehensive analysis model, and grey classification can then be carried out to achieve comprehensive evaluation of the target object, which makes the evaluation work more quantitative and intuitive.

**Author Contributions:** Conceptualization, J.L. and X.C. Methodology, J.L. and X.C. Data Curation, J.L. Writing—Original Draft Preparation, J.L. Writing—Review and Editing, J.L., X.C., C.G. and Z.H.

**Funding:** This research was funded by the National Natural Science Foundation of China under Grant No. 51609217 and Grant No. 51679222.

**Acknowledgments:** Appreciation is due to all members of Li Zongkun's research group of Zhengzhou University for their valuable opinions on this study.

**Conflicts of Interest:** The authors declare no conflict of interest.

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
