# Peer review of "Seepage Comprehensive Evaluation of Concrete Dam Based on Grey Cluster Analysis"

_water, doi:10.3390/w11071499_

Round 1
Reviewer 1 Report
This study proposed comprehensive indication system using mathematical model. It was however afraid that some factors regard to phenomena or behavior for materials described with enough proven. It is so large problems for articles. This paper addressed in simulation mathematical calculation models and it is slight to corresponding into practice behaviours.
Line 33
Please would you like to apply more information regard to damages induced by temperature into dam.
Line 26
Please would you like to mention practice problems related to purpose of this study with detail.
Line 132
Creep of the rock is so interesting that this study must indicate engineering problems in deformation of dam.
Line 85
Authors focused some factors that they are interesting in practice. They were however only assumptions and experimental data sets were NO mentioned. It should be appear with considering comments from literatures.
Line 137
The parameters were indicated that please would you like to conceptual reasons.
Line 279
Authors take into account for calculated data sets that there were lack essential materials phenomena in discussion processes.
Reviewer 2 Report
The paper is focused on the seepage evaluation of concrete dams based on multi-indicator grey clustering analysis model. At the beginning, a brief introduction of the problem is given, and then the modeling steps are analyzed. The paper reports also the results obtained with an application of the procedure considered on a real case study, a concrete dam located in China.
It is opinion of the Reviewer that the paper may be published, after some revisions. Before its publication the following major reviews should be carefully taken into consideration:
· It is not clear throughout the manuscript, which is the novelty of the work presented. The Authors should carefully remark, for example, the current state of art and what’s new in the paper presented. It is not clear, if the results shown are merely derived from an application of an already-known method or, else, there is a new procedure proposed. Please, clarify this.
· It is known that the seepage is also correlated to characteristics of the dam material. Why has not this aspect been taken into account in the paper? Please, remark also how this aspect may be implemented in the proposed procedure. If available, please insert a brief description of the dam concrete material of the considered case study.
· The section 4.1. should better describe how the seepage monitoring model has been derived starting from the model output values.
· In the conclusion should be remarked how the results obtained by applying the procedure discussed may be useful for the dam maintenance. This for sure would make the presented work much ore fashionable also in other engineering fields, such as for instance, the structural one.
Round 2
Reviewer 2 Report
The paper has been improved in according to the comments provided. It is opinion of the Reviewer that it may be published in the present form